# Recent Advances in Biomimetic Nanocarrier-Based Photothermal Therapy for Cancer Treatment

**DOI:** 10.3390/ijms242015484

**Published:** 2023-10-23

**Authors:** Juan Gallo, Aranzazu Villasante

**Affiliations:** 1Advanced Magnetic Theranostic Nanostructures Lab, International Iberian Nanotechnology Laboratory (INL), 4715-330 Braga, Portugal; juan.gallo@inl.int; 2Nanobioengineering Lab, Institute for Bioengineering of Catalonia (IBEC), The Barcelona Institute of Science and Technology (BIST), 08028 Barcelona, Spain; 3Department of Electronic and Biomedical Engineering, Faculty of Physics, University of Barcelona, 08028 Barcelona, Spain

**Keywords:** photothermal therapy, biomimetic nanoparticles, membrane-camouflaged nanoparticles, cancer treatment

## Abstract

Nanomedicine presents innovative solutions for cancer treatment, including photothermal therapy (PTT). PTT centers on the design of photoactivatable nanoparticles capable of absorbing non-toxic near-infrared light, generating heat within target cells to induce cell death. The successful transition from benchside to bedside application of PTT critically depends on the core properties of nanoparticles responsible for converting light into heat and the surface properties for precise cell-specific targeting. Precisely targeting the intended cells remains a primary challenge in PTT. In recent years, a groundbreaking approach has emerged to address this challenge by functionalizing nanocarriers and enhancing cell targeting. This strategy involves the creation of biomimetic nanoparticles that combine desired biocompatibility properties with the immune evasion mechanisms of natural materials. This review comprehensively outlines various strategies for designing biomimetic photoactivatable nanocarriers for PTT, with a primary focus on its application in cancer therapy. Additionally, we shed light on the hurdles involved in translating PTT from research to clinical practice, along with an overview of current clinical applications.

## 1. Introduction

### 1.1. Photothermal Therapy for Cancer Treatment

Cancer is still one of the most significant challenges of the twenty-first century [1]. The complexity of the condition (many diseases encompassed in one name) and the high heterogeneity of individual tumors themselves contribute to the difficulty of this challenge [2]. Immunotherapy has recently joined the three classic pillars of clinical cancer treatment (surgery, chemotherapy, and radiotherapy), bringing new hope to oncological patients [3]. Moreover, many other treatment modalities are getting increasingly closer to clinical application. This selection of different therapeutic modalities is needed due to cancer’s complexity. Within these new modalities, one that has already reached the clinic in some form is hyperthermia [4].

Heating up cancer tissues can treat the disease due to the increased susceptibility of tumor cells to temperature. On top of that, a T increase in the tumor microenvironment (TME) can facilitate the penetration of drugs and medical formulations deep into solid tumors. Photothermal therapy (PTT) is one of the most promising versions of hyperthermia due to its safety and high externally controlled specificity. In PTT, a material (photothermal agent, PTA) acts as an intermediary between a radiation source and the desired toxic effect on cancer cells by converting electromagnetic radiation into heat. In terms of the excitation source, near-infrared (NIR) wavelength is preferred, mainly for three reasons: (i) longer wavelength radiation can penetrate deeper into biological tissues; (ii) NIR falls within what is known as biological windows (wavelengths where the absorption by biomolecules is reduced); and (iii) NIR radiation energy is not high enough to induce biochemical changes on healthy cells and thus is safer than, e.g., UV radiation (able to damage DNA) [5].

The high specificity and safety of PTT rely on the two-component nature of the treatment: radiation source and photothermal agent. The exposure of a tissue to (NIR) laser radiation is not enough to generate a temperature change on its own. In the same way, the presence of photothermal agents should not produce any toxic effect on its own. This two-component nature is of the utmost importance as it allows us to surpass some limitations of irradiation systems and nanoparticle-based drugs. On the side of the irradiation, as the electromagnetic radiation passes through healthy tissues on its way to the tumor, it will not exert any deleterious effect on those tissues as long as no PTA is present. Modern laser systems provide high control over the irradiated area and stability of the radiation, spectral width, and power output. On the PTA side, many of the most successful PTAs are nanoparticle-based (see below), and as such, they tend to accumulate in ‘filtering’ organs like the liver, the spleen, or the lungs. These organs will not suffer any deleterious effect from the PTAs as long as they are not exposed to the laser. Overall, all these characteristics make PTT a particular and safe treatment modality. In any case, the delivery of PTAs to the tumor site in a dose that is high enough to induce a significant temperature change when irradiated is still one of the main bottlenecks of this therapy and nanomedicine in general. 

Inorganic materials tend to be cleared from circulation by the mononuclear phagocyte system, while organic molecules are rapidly excreted via the kidneys and frequently present solubility issues. Targeted strategies based on functionalizing these materials with specific antibodies, peptides, or aptamers have only improved tumor accumulation modestly [6]. In this scenario, biomimetic strategies using cell membranes and natural structures to coat PTAs are being heavily explored to improve circulation time and tumor accumulation. In this review, we present an overview of the most promising PTAs, together with the most current biomimetic strategies, to increase the efficiency of PTT. We also give some selected examples of PTT applications and discuss the limitations and prospects of the technique.

### 1.2. Nanocarriers for Photothermal Therapy

As mentioned above, PTT relies on localized electromagnetic energy conversion into heat. Speaking in general, the outcome of the PTT process from the side of the photothermal agent depends on parameters inherent to the photothermal agent’s nature (mainly the photothermal conversion efficiency) plus effects that are more dependent on physicochemical parameters and/or related to the functionalization of these materials (e.g., specificity, cell penetration abilities) [7]. Photothermal conversion efficiency (PCE) has been extensively studied on Au nanostructures. For these particular plasmonic materials, the PCE depends mainly on the plasmon resonance wavelength, particle volume/size (PCE of Au nanoshells decreases with increasing size) [8], shell coating, and assembly [9]. For semiconductor materials, the band gap also plays a key role. Optimizing these materials is a lengthy process in which a compromise must be reached between PCE, synthesis complexity, functionalization, and biocompatibility/safety.

Since the introduction of nanotechnology into biomedicine, particularly oncology, the number of therapeutic options reported at a research level has increased significantly. This is also true in the area of PTT. There is currently a comprehensive battery of material options described as PTT agents. Among them, Au nanomaterials and polydopamine (PDA) nanostructures are arguably the most important. Nevertheless, others have shown excellent properties as PTT agents beyond these two materials. For example, carbon-based materials, either carbon nanotubes (CNTs) or graphene, have been widely studied in this field. Recently, semiconductor materials such as CuS, MoS_2_, Bi_2_S_3_, Bi_2_Se_3_, and black phosphorus have been introduced. Other materials already proposed for different applications in biomedicine (for example, Fe and Mn oxides, mainly used as magnetic resonance imaging contrast agents) have also shown promise in this field. Finally, other organic materials apart from PDA have been proposed as PTT agents in nanoparticle form, such as polypyrrole. In the following section, we will describe these materials in more detail.

Once NPs are administered to a patient, biomolecules in the blood serum spontaneously form adsorption layers around the NPs called protein corona, which leads to recognition and faster clearance by the immune system. As a result, nanoparticle-targeting capabilities are profoundly reduced.

Coating the NPs’ core with an outer shell of different materials has evolved as an excellent strategy to obtain nanoparticles that can escape or attenuate the immune response. Among different approaches to shell fabrication, functionalization with biomolecules has been demonstrated to increase NPs’ stability in biological fluids and improve specific cancer targeting. The use of simple molecules, such as PEG, to modify the surface of the particles has been shown to reduce aggregation and lower protein corona formation. However, PEG can induce a specific immune response, producing anti-PEG antibodies that facilitate PEGylated nanoparticle phagocytosis. Another popular shell is made of antibodies that specifically recognize cancer surface proteins. However, antibodies on the NP shell influence corona generation, which activates the immune system and enhances the immune elimination of NPs from the bloodstream and tissues. Moreover, using one specific antibody or targeting ligand is not always effective because of the phenotypic heterogeneity of tumors, which also decreases the targeting of the tumor bulk and treatment efficacy.

Since traditional shells show significant limitations, innovative approaches based on the generation of biomimetic nanoparticles have emerged in the last few years. Biomimetic nanoparticles combine the desirable biocompatibility properties and immune evasion of natural entities (i.e., cell membranes, extracellular vesicles, toxins, and pathogens) and the tunable capacity of synthetic nanoparticles. We discuss the two main strategies based on biomimetic nanoparticles for PTT that have demonstrated efficacy in specific cancer targeting and avoiding immune clearance, namely membrane-camouflaged nanoparticles and nature-inspired nanoparticles. Additionally, we highlight how the different NIR-responsive materials discussed above can be employed as cores for nanoparticles, applying different shells and coating approaches (Figure 1).

## 2. Core: NIR-Responsive Materials

As mentioned briefly above, many materials with good PCE properties have been described as PTT agents. In this section, rather than collating a comprehensive collection of these materials, we will present the different families of materials currently being investigated as PTT agents. References to relevant review works on each type of material will be provided for those eager to delve into the details of each family.

### 2.1. Organic Materials

Two of the most commonly used materials for PTT fall within this category, polydopamine (PDA) and indocyanine green (ICG), together with other less-known materials such as polypyrrole or porphyrins (better known for their photodynamic properties). While PDA is used in the nanoparticle form (either as the primary or sole component of the nanostructure or as a coating on a nanoparticle of different nature) [10,11], ICG is frequently used as cargo within a nanostructured drug delivery system [12,13]. The main advantage of ICG is that it is already approved for human use for the diagnosis of cardiac, circulatory, hepatic, and ophthalmic conditions (“Definition of indocyanine green solution—NCI Drug Dictionary—NCI”, 2011) [14]; as such, a potential translation from bench to bedside is expected to be easier than other options. A covalent functionalization strategy would be preferred as it provides closer control over the fate of the dye. However, direct coupling of native ICG is chemically challenging, and thus, encapsulation/entrapment is the preferred option. ICG derivatives have been developed to solve this reactivity issue (e.g., IR808, IR825, IR2), but these structures lose the ease of translation advantage.

Regarding PDA, a bottom-up approach is usually followed for its preparation. Although several synthetic methods are available (enzymatic, electropolymerization) [11,15], the in situ oxidation of dopamine (DA) in solution is the most commonly used technique due to its simplicity. The mechanism of this polymerization is still controversial, although two main pathways seem to co-exist: oxidative polymerization and physical self-assembly [16]. This reaction is used for the production of PDA NPs to add a PDA coating on top of NPs of a different nature (the ability of DA to adhere to a plethora of different materials through either the amine or the catechol moiety aids the versatility of this approach) and for the production of PDA nanocapsules (usually by coating an intermediate NP as a template that is later selectively digested). A (covalent) conjugation strategy is also used for some applications in the case of the PDA coating on different nanomaterials. PDA can react via Schiff base or Michel addition reactions with amines or thiols [17]. The size of the final NPs can be controlled by controlling parameters such as the reaction temperature and pH or DA concentration.

### 2.2. Gold (Au) Nanomaterials

Au nanomaterials are the best-studied inorganic nanomaterials for PTT applications. Relatively large nanostructures are needed for efficient PTT (nanorods, nanocapsules, nanostars) to increase NIR absorption by matching the laser wavelength and the Au nanomaterial surface plasmon. The absorption wavelength of the surface plasmon depends on different parameters depending on the structure of the Au nanomaterial. For example, in the case of nanorods, the aspect ratio is the parameter that controls the plasmon resonance. In the case of nanocapsules, the plasmon depends on the size and thickness of the Au shell, while in the case of nanostars, the plasmon depends on the size and thickness of the tip of these anisotropic materials [18]. The size/shape also controls the PCE of Au nanomaterials [19]. Smaller diameters generally lead to a higher PCE, while anisotropic shapes increase the PCE by increasing the absorption cross section [20]. Besides their excellent PCE properties, Au nanomaterials present very high stability, low reactivity, well-established functionalization chemistry, and good biocompatibility, making them appealing materials for biomedical applications. Apart from these properties, the popularity of Au nanostructures as PTT agents comes from the precise control available over the synthesis and final physicochemical (size, shape, etc.) properties of the nanostructures. Protocols are available for preparing Au nanostructures of different sizes and shapes, most of them being relatively simple. The most common protocol for preparing nanorods and nanostars is two-step seeded growth, where spherical nanoparticles are synthesized first and then used as seeds to grow the final structure. In this protocol, the growth reaction time and the Au precursor concentration control the nanostructures’ final size and aspect ratio [21]. Template-directed synthesis is the most common protocol for preparing nanocapsules as it also is similar to creating PDA coatings and shells [22].

### 2.3. Carbon-Based Nanomaterials

#### 2.3.1. Carbon Nanotubes (CNTs)

CNTs are one of the most iconic structures of nanotechnology. These structures have been extensively studied and investigated for various applications, including biomedicine. PTT can be a target application for CNTs due to their strong light absorption capability in a wide wavelength range, including NIR and high PCE. CNTs are cylindrical graphite sp^2^ sheets that can be classified into single-walled CNTs (SWCNTs) and multi-walled CNTs (MWCNTs), depending on the number of concentric graphite layers. The optical properties of these materials depend on their diameter and chirality (relative orientation of the carbon hexagons concerning the tube axis); this last property is difficult to control with most synthetic protocols for SWCNTs, resulting in absorption spectra for these samples with several absorption peaks superimposed [23]. MWCNTs present simpler absorption spectra where the absorption is inversely proportional to the wavelength. Despite these absorption differences, both nanostructures have been widely investigated as PTT agents [24,25].

#### 2.3.2. Graphene

Closely related to CNTs, graphene and graphene family (graphene oxide and reduced graphene oxide) nanomaterials are also being heavily explored as PTT agents. These materials also present a sp^2^ carbon structure. Graphene oxide (GO) is probably the most explored member of this family. GO is commonly prepared through chemical oxidation (e.g., Hummer’s method and modifications) [26] and the exfoliation of graphite. GO is also the precursor for preparing reduced GO (rGO) through the exposure of GO to reducing agents at relatively high temperatures. In rGO, the graphitic lattice is more conserved than in GO, which enhances the NIR absorption of the material [27]. However, as with CNTs, both GO and rGO have been proposed for PTT applications [28].

### 2.4. Other (Inorganic) Nanomaterials

Two well-differentiated groups of materials can be found within this category. First, many semiconducting materials (copper-, bismuth-, and tungsten-chalcogenides; MoS_2_; black P; Ag_2_S; WO_3_, etc.) have been proposed as PTT agents. This group generally presents excellent NIR absorption properties and PCE, but their long-term biocompatibility is still mostly untested. The other group is formed by oxide materials, both Fe- and Mn-based. The PTT performance of these materials is frequently worse than that of the other group, coming from their worse intrinsic PCE. However, their lack of performance is compensated by the fact that these materials have been widely studied for biomedical applications (some of them, like Fe oxides, are already approved for their use in humans), and their safety and biocompatibility are assumed to be high.

#### 2.4.1. Semiconductor Materials

Many of the members of this family are 2D materials (black P, MoS_2_, WS_2_, Bi_2_X_3_, WO_3_), similar in structure to graphene. As in the case of graphene, their preparation is possible through a *top-down* approach via chemical exfoliation. However, this protocol is usually time consuming, and the physicochemical properties of the final nanomaterials are, in most cases, difficult to control. Thus, *bottom-up* protocols have been developed to prepare most of these materials [29]. For example, solvothermal reactions have been developed to prepare MoS_2_ nanosheets [30,31]. Within the chalcogenide compounds (MoX_2_, WX_2_, Bi_2_X_3_, CuX), S, Se, and Te compounds have been researched; although in many cases the PCE of the Se and Te compounds is outstanding, the S version of these materials is by far the most studied due to a combination of raw S materials availability, perceived biocompatibility, and simpler synthetic protocols [32,33]. Combinations of the above materials have also been researched for PTT (e.g., Bi_2_S_3_-MoS_2_) [34].

#### 2.4.2. Oxide Materials

Fe and Mn oxides are versatile materials with a high multifunctional potential in the biomedical arena. Depending on their structure (many oxides are described for Fe and Mn with very different properties; γ-Fe_2_O_3_, α-Fe_2_O_3_, Fe_3_O_4_, MnO, MnO_2_, Mn_3_O_4_), Fe and Mn oxides can be used as contrast agents in magnetic resonance imaging (MRI) [35,36], as magnetic hyperthermia effectors [37], for magnetic guiding purposes [38], as cytotoxic probes through Fenton and Fenton-like reactions [39], and as O_2_ generating structures [40]; in addition, as mentioned above, Fe oxide nanoparticles (magnetite and maghemite) reached the clinic in the 1990s as MRI contrast agents and are still used worldwide as iron supplements. Altogether, these properties justify applying these oxide materials as PTT agents. Beyond the application potential, the availability of a wide variety of protocols for preparing these oxide nanomaterials, covering the whole spectra of complexity and quality requirements, also contributes to the popularity of these structures [41,42].

## 3. Biomimetic Shells

### 3.1. Membrane-Camouflaged Nanoparticles

Recently, cell-derived membranes have been used to cover nanoparticles with external decoration to escape immune recognition and improve their blood circulation time. This process consists of the complete transfer of the native markers present on cell membranes or extracellular vesicles (EVs) of source cells onto the surface of the different nanomaterials. Different cell sources can be used to cloak nanoparticles using different coating approaches. This section will only focus on eukaryote-derived membranes.

#### 3.1.1. Cell-Membrane-Cloaked Nanoparticles

Cell-membrane-camouflaged NPs are a top-down approach to coat synthetic NP cores with natural cellular membranes to avoid immune clearance and increase circulation time in the body, specificity, and efficacy. Researchers have camouflaged NPs with cell membranes from healthy cells (i.e., red blood cells, platelets, macrophages, MSC, fibroblasts) and cancer cells. Both cell sources have demonstrated their unique ability to increase the immune system’s shielding effect of protein corona and reduce phagocytosis. For example, coating NPs with erythrocyte membranes enables a prolonged blood half-life, while platelet membrane and mesenchymal cell coating allow immune evasion and specific homing to tumor tissues. Notably, using cancer cell membranes increases anti-cancer treatments’ targeting specificity and efficacy.

##### Healthy Cell Membranes

The immune system can recognize “markers of self” on the surface of healthy cells, such as the CD47 protein on Erythrocytes (red blood cells, RBCs), which can bypass the clearance by macrophage engulfment. Effectively escaping immune recognition in the body also benefits prolonged blood circulation. Thus, researchers have taken advantage of the effective immune recognition in healthy cells to use them as natural carriers (Table 1).

Red blood cells—The application of healthy cells as vehicles of bioactive molecules was initially studied with RBCs. In 1953, Gardos reported the first encapsulation of a molecule into RBCs. However, Jain’s “carrier red blood cells” concept was not established until 1979 to describe drug-loaded erythrocytes [60]. In 2011, Hu and co-workers proposed for the first time a top-down biomimetic approach to particle functionalization by fusing, using an extrusion approach, RBC membrane vesicles onto poly(lactic-*co*-glycolic acid) (PLGA) nanoparticles [61]

Since then, RBC-membrane-camouflaged nanoparticles have been strongly verified as promising candidates for cancer therapy. 

Gold was the first photothermal conversion material to be cloaked with a cellular membrane, specifically, RBC membranes [62]. Although this study showed how solid gold nanospheres could be coated entirely with RBC vesicles to evade macrophage uptake, researchers did not further analyze the photothermal effect of AuNP@RBC on cancer cells. However, this first report laid the foundation for subsequent coating of NIR-responsive nanomaterials with RBC membranes for PTT, opening a new area of anti-cancer strategies [43,44]. Thus, Piao et al. reported the fabrication of RBC-membrane-coated gold nanocages (RBC-AuNCs) for studies of PTT [44]. They demonstrated the tumor uptake of NPs and cancer cell ablation via photothermal effects in vitro and in mice. However, unsurprisingly, RBC-AuNCs showed the highest accumulations within the liver and spleen, not in tumors.

Besides gold, iron-oxide-based nanomaterials have been reported recently and are raising great interest for tumor PTT. For example, iron oxide magnetic nanoclusters (MNC) coated with RBC membranes (MNC@RBCs) demonstrated prolonged blood circulation time and significantly enhanced PTT efficacy. The treatment with MNC@RBCs and NIR light significantly reduced the average tumor weight using a breast cancer xenograft mouse model [45]. To increase the photothermal conversion efficacy of MNC@RBCs, Wang and co-workers loaded NIR cypate molecules to the MNC (Cyp-MNC@RBCs) [46]. The treatment with cyp-MNC@RBCs plus laser irradiation, compared to their MNC@RBCs counterpart, significantly inhibited tumor growth in HCT-116 tumor-bearing mice.

Current efforts to further improve NP biodistribution and tumor homing include using different NIR-responsive nanomaterials beyond gold and iron oxide and functionalizing RBC membranes with active targeting ligands. In addition, PTT is often combined with chemotherapy, and NPs are co-loaded with chemotherapeutic drugs to enhance the anti-cancer effect [47,48,49,50]. For example, Luo and co-workers showed the use of PEGylated graphene oxide quantum dots to co-load indocyanine green (ICG) as the PTA and doxorubicin (DOX) (a broad-spectrum anti-cancer drug) for therapy of cervical cancer. ICG-DOX-NPs were “camouflaged” with RBC membranes to avoid immune clearance; they were also decorated with folic acid to endow the targeting ability as cervical tumor cells express high levels of the folate receptor [49]. They demonstrated that the presence of folic acid increased the accumulation of NP at the tumor site in HeLa-tumor-bearing mice. Luo et al. showed that NIR laser irradiation of the NPs, besides inducing abundant ROS, also significantly increased the entry of DOX into the cell nucleus.

Another example of the combinational strategy is Prussian blue NPs loaded with gamabufotalin (CS-6) and cloaked with hyaluronic-acid-decorated RBC membranes [50]. Hyaluronic acid can bind specifically to the CD44 receptor of breast cancer cells, while Prussian blue acts as a PTA for PTT, and CS-6 is an anti-tumor drug that inhibits VEGF-induced angiogenesis in vitro, on cancer cells, and in vivo on mouse tumor xenografts. The combination of these three components increased the circulation time of NPs in the blood and the ability to escape from immune surveillance, as previously discussed for other RBC-cloaked nanoparticles. In addition, modification with HA enables targeted accumulation of CS-6 at tumor sites.

Platelets—Platelet (PLT)-membrane-camouflaged nanoparticles have also been widely explored for cancer therapy. It has been shown that coating with platelet membrane increases evasion of immune clearance while favoring interactions with damaged vasculature and tumor tissues [63]. For example, mesoporous silica-coated bismuth nanorods camouflaged by platelet membrane (BMSNR@PM) escaped the immune system by lowering macrophage endocytosis. Also, treatment with BMSNR@PM significantly lowered the tumor size of 4T1-tumor-bearing [51]. Importantly, different studies have demonstrated that PLT can recognize and interact with circulating tumor cells (CTCs) [64], which points to PLT membranes as candidates for specific targeting of CTCs.

As occurs for RBM-camouflaged NPs, it is frequent to combine PTT with chemotherapy. Due to its efficacy in fighting many cancers, doxorubicin (DOX) is the most common anti-cancer drug used as a proof of concept. Wu et al. fabricated polypyrrole (PPy) nanoparticles (as photothermal agents) encapsulating DOX that were cloaked with PLT membranes [52]. The photo-chemotherapy based on PLT-PPy–DOX nanoparticles effectively suppresses primary tumor growth and inhibits tumor-distant-metastasis in an orthotopic mouse model of hepatocellular carcinoma. Another DOX example is PLGA-loaded nanoparticles combined with the NIR fluorescent dye IR780 for chemotherapy and PTT [53]. The IR780@PLGA/Dox nanoparticles camouflaged with PLT membranes showed longer retention times in the bloodstream of breast cancer 4T1-tumor-bearing Balb/c mice, which also accumulated at the tumor site. 

Ferroptosis is a new type of programmed cell death discovered in 2012 that is mediated by iron-dependent Fenton-like reactions [65]. Iron accumulation leads to the continuous induction of lipid peroxidation, considered the leading killer in ferroptosis. Besides their use for PTT, iron oxide NPs can induce ferroptosis and are emerging as promising candidates for combinational anti-cancer therapy.

The work published by Jiang et al. is an exciting example of using PLT membrane-camouflaged Fe_3_O_4_ NPs to induce ferroptosis [54] (Figure 2). The authors based their strategy on sulfasalazine (SAS)-loaded iron oxide (Fe_3_O_4_) nanoparticles coated with PLT membrane (Fe_3_O_4_-SAS@PLT) (Figure 2A). They showed that Fe_3_O_4_-SAS@PLT NPs inhibited the viability of triple-negative mouse breast cancer cells (4T1 cells) (Figure 2B) and presented enhanced cellular uptake as compared to those without coating (Figure 2C). The authors further demonstrated that the mechanism of cell death via ferroptosis was inhibiting the glutamate-cystine antiporter system Xc−transporter pathway. They also showed the efficacy of Fe_3_O_4_-SAS@PLT in vivo using a mice metastatic model generated through intravenous injection of 4T1 cells expressing firefly luciferase (4T1-Luc). Bioluminescence imaging confirmed that tumor cells had spread to the lung on day 7 after cell injection (Figure 2D). Then, the mice were treated with DiD-labeled Fe_3_O_4_-SAS@PLT, and the fluorescence signal of the NPs was recorded. They observed the ability of Fe_3_O_4_-SAS@PLT for homing to 4T1 metastatic tumors (Figure 2D,E). They concluded that the Fe_3_O_4_-SAS@PLT-mediated ferroptosis and anti-PD-1 immunotherapy effectively inhibited tumor growth (Figure 2F).

Macrophages—The Macrophage (Ma) membrane is also ideal for camouflaging NPs. Numerous studies have taken advantage of such biomimetic derivatives to increase biocompatibility, blood circulation, tumor accumulation, and decrease opsonization [55].

Amongst NIR-responsive materials, iron oxide and gold have been the preferred option to be camouflaged with macrophage membranes. An Fe_3_O_4_ core plus the macrophage membrane shell (Fe_3_O_4_@Ma) exhibit good biocompatibility, immune evasion, cancer targeting, and light-to-heat conversion capabilities [55]. In addition, Fe_3_O_4_@Ma NPs were shown to have a therapeutic effect in in vivo studies. BALB/c mice bearing MCF-7 tumor xenografts treated with Fe_3_O_4_@Ma NPs exhibited dramatic tumor regression over time [55]. Gold-nanoshell-coated mesoporous silica nanoparticles can also be cloaked with macrophage membranes, effectively reducing tumor growth on 4T1-tumor-bearing mice upon NIR irradiation [56]. In a similar work, Madsen and co-workers reported gold–silica Ma-membrane–camouflaged nanoshells (AuNS) for PTT of rat gliomas [57]. Gliomas are the most frequently occurring brain tumors supported by tumor-associated macrophages (TAMs), among other cells. They are recruited from peripheral blood to the tumor, bypassing the blood–brain barrier, which usually avoids the penetration of anti-cancer agents inside the brain. Thus, Madsen and co-workers took advantage of the innate tropism of macrophages to migrate to the glioma environment for developing Ma-membrane-camouflaged NP. Their strategy involved incubating PEGylated AuNS (a 120-nm silica core with a 12–15-nm gold shell) for 24 h with already seeded rat alveolar macrophages. Then, they detached and selected AuNS-loaded macrophages for in vitro and in vivo studies. Therefore, this approach differs from the mentioned above as they use entire living macrophages instead of only membranes. It is a Trojan horse strategy rather than a coating. The authors proved the effectiveness of the PTT using a Sprague–Dawley rat glioma model, where rats received a direct intracranial injection of C6 glioma cells.

Mesenchymal stem cells—Similar to TAMs, mesenchymal stem cells (MSC) have demonstrated migration towards tumor microenvironments and immunogenicity. Only a few works have been reported using MSC membranes for “camouflaging” NPs, and most of them used polydopamine (PDA) as the core. PDA is a bioinspired material with exciting features, simple preparation protocols and functionalization procedures, biocompatibility, free radical scavenging, and photothermal/photoacoustic properties [66]. Zhang and co-workers used PDA NPs coated with MSC membrane and encapsulated SN38, an antineoplasic drug (PDA-SN38@SCM), to treat bone tumors [58]. In this case, they first prepared MSC-derived vesicles by cell lysis and physically extruding the cell membrane through a 400 nm polycarbonate membrane. Then, polydopamine nanoparticles (PDA-NP) were mixed with MSC-derived vesicles and sonicated. Finally, SN38 was loaded into the PDA@MSC particles. This is, again, an exciting strategy to combine multiple therapies with different modes of action. Zhang’s work demonstrated low toxicity and efficacy in tumor treatment and synergistic chemo-photothermal therapy.

Loading NIR-responsive materials with small interfering RNA (siRNA) is another attractive combinatorial strategy for cancer treatment. siRNAs are double-stranded RNAs that target complementary mRNAs to suppress the expression of specific genes, including those that encode for cancer (knockout function). PDA-Fe_3_O_4_ NPs coated with mesenchymal stem cell membranes were demonstrated to be an excellent carrier for Plk1-siRNA with good photothermal conversion capability in vivo [59]. Additionally, these NPs show an anti-tumor effect when injected in DU145 xenograft mice after NIR laser irradiation [59].

##### Cancer Cell Membranes

Although technological success has been achieved in NP functionalization, one of the main obstacles is still the precise cancer homing. Cancer cell membranes are becoming a promising and efficient approach to overcome that issue. Cancer cells can recognize each other and adhere to one another through a mechanism known as homotypic binding [67]. The functionalization of nanoparticles with cancer cell membranes aids homotypic targeting by source cells because of its self-recognition property (Table 2).

Currently, strategies for developing effective cancer membrane-coated NPs are the same as those using healthy cell membranes. Researchers usually choose a NIR-responsive material as a core (mainly iron oxide, gold, CuS, or PDA) or mesoporous silica with a PTA with NIR fluorescence, such as ICG. PLGA has also been exploited in many NPs as the core in combination with ICG. Then, depending on the type of cancer to target, the selection of the outer shell will differ [68,69,70,71,72,73,74,75]. For example, Fe_3_O_4_ NPs coated with silicon dioxide (SiO_2_) can be camouflaged with HepG2 hepatoma-cell-derived cellular membranes for enhancing NK-cell-based immunotherapy [68]. The functionalized nanoparticles carry cancer-specific antigens on the surface that activate NK cells. This strategy enables upregulation of the expression of surface activating receptors on NK cells and subsequent secretion of cytotoxic factors for enhancing NK-cell-mediated anti-tumor effects against HepG2 cells. 

Li and co-workers published another interesting example. They designed a PLGA NP encapsulating perfluorocarbons (PFCs) with ICG and coated with the human lung A549 cancer cell membrane (AM-PP@ICGNPs) [69]. The cancer cell membrane modification could effectively enhance the circulation times of NPs by 1.7 times compared to NPs without membrane coating and 3 times to free ICG molecules. Also, AM-PP@ICGNPs showed excellent photothermal conversion efficiency, and with the homologous targeting ability, AM-PP@ICGNPs accumulated mainly in the tumor site in A549 xenografted tumor mice. Importantly, tumor growth inhibition was achieved in AM-PP@ICGNP-treated mice after mild localized laser irradiation (765 nm, 400 mW/cm^2^, 15 min), which confirms the great potential of photothermal therapy. 

Although PTT alone is reaching exciting results, the synergistic effect of PTT and chemotherapy is currently desirable when cloaking the NP with cancer cell membranes. Combining PTT with chemotherapy offers many advantages versus either treatment alone. For example, a nano-drug delivery platform was constructed by liposome vesicles containing the PTA ICG and doxorubicin as a chemotherapy drug and coated with hepatocellular carcinoma HepG2 cell membrane (ICG-Dox-HepM-TSL) [71]. ICG-Dox-HepM-TSL nanoparticles can escape the immune system and precisely target the nude mice tumor sites bearing recurrent HepG2 tumors. In another example, ICG and decitabine (DCT) were co-encapsulated in PLGA NPs and coated with 4T1 breast cancer cell membranes to produce the biomimetic nanoparticle 4T1-PLGA@ICG/DCT (Figure 3) [74].

DCT is a DNA methylation inhibitor used to activate the gasdermin E (*GSDME*) gene, which is frequently methylated and silenced in most tumor cells. When the *GSDME* gene is demethylated and the protein can be expressed, this can be cleaved by activated caspase-3, which leads to a type of programmed cell death known as pyroptosis accompanied by inflammatory and immune responses [76]. The 4T1-PLGA@ICG/DCT nanoparticle combines chemical and photo treatments to synergistically induce cell pyroptosis so that NIR laser irradiation produces at the same time local hyperthermia, caspase-3 activation and DCT release (Figure 3). Pyroptotic cancer cells secrete inflammatory factors that activate anti-tumor immunity to inhibit primary and distant tumor growth in 4T1-tumor-bearing mice (Figure 3).

##### Hybrid Cell Membranes

An important strategy that has emerged in the last few years is using hybrid cell membranes. The advantage of this approach overcoating with a single cell type membrane is the presence of specific proteins of both cell types, which should combine the functionalities from both cell membranes. For instance, the macrophage membrane presence helps NP avoid opsonization, while cancer cell membranes favor homotypic targeting. Therefore, combining both should improve NPs’ precise cancer homing and reduce immune clearance. In one example, photo-chemotherapy based on macrophage (RAW 264.7 cells)–cancer cell (H22) hybrid membrane-coated hollow copper sulfide NPs encapsulating sorafenib and surface modified with anti-VEGFR (CuS-SF@CMV NPs) was developed [73]. Sorafenib is a multi-kinase inhibitor that targets both MEK and VEGFR, and its effects were further enhanced by the inclusion of the anti-VEGFR antibody on the surface of CuS-SF@CMV NPs. The ~10 nm thick RAW 264.7-H22 uniform hybrid membrane contained specific membrane proteins of both cell types, such as the macrophage-specific CD135 (Fms-like tyrosine kinase 3) and the E-cadherin from the H22 cell. The authors reported that the presence of the hybrid membrane significantly increased the ability of the CuS@CM NPs to localize to the homotypic cells and escape the immune cells. Also, the in vivo murine hepatoma model treated with CuS@CM NPs and NIR irradiation exhibited smaller tumors than the control counterparts, confirming this approach’s potential therapeutic effect.

Bacterial-cancer cell-membrane-coated nanoparticles are another compelling example of using hybrid cell membranes. One advantage of using bacterial membranes is that the bacterial outer membrane presents immune activation properties. The outer membrane can trigger the production of anti-tumor cytokines by immune cells and induce the subsequent immune response for immunotherapy. Wang et al. [75] reported upon the preparation of hollow polydopamine (HPDA) NPs covered by the fusion of a bacterial outer membrane vesicle (OMV) and B16-F10 cancer cell (CC) membrane to obtain HPDA@[OMV-CC] NPs. The NPs intravenously injected into the C57BL/6 melanoma-bearing mice showed particular accumulation in the liver, lung, tumor, and lymph node of the mice treated. The tumor progression was significantly inhibited after exposure to NIR laser irradiation, presenting a tumor growth inhibition (TGI) rate of about 99.9%. Notably, the TGI for mice treated with HPDA@[OMV-CC] but not irradiated was 31.7%, demonstrating that the fused OMVs induced immunotherapy to inhibit tumor growth. However, this strategy can be risky as immune activation will occur unspecifically wherever the NPs accumulate, and severe off-target effects can be expected. We will further discuss bacterial-membrane-coated NPs in Section 3.2.2. Bacteria-derived nanoparticles. 

#### 3.1.2. Cell-Derived Extracellular Vesicles

Recently, the use of extracellular vesicles (EVs), such as exosomes, as the natural material of biomimetic NPs is becoming of great interest. EVs are small membrane vesicles secreted by all types of cells that have an essential role in cell–cell communication. To reach the target sites and be internalized by the recipient cell, EVs present ligands in their surface membrane that bind to specific receptors in the host cells. The nature of ligands depends on the cells from which they originated [77,78]. Due to these surface ligands, EVs display a high targeting specificity and efficient cellular uptake. Considering EVs’ natural targeting capabilities and low immunogenicity, they are emerging as an ideal surface coating for preparing biomimetic NPs (Table 3).

Different methods for encapsulating nanoparticles in EVs have been reported, such as passive incubation, freeze/thaw cycling, surface conjugation, extrusion, electroporation, and sonication [85,86] Recently, labeling parental cells has appeared as an emerging technique for the fabrication of EV-coated NPs. This method involves direct incubation of parental cells with NPs, cell internalization, incorporation of the NPs into the exosome biogenesis pathway, and exocytosis of NP-loaded EVs to the culture medium [86]. However, the efficacy of this approach is challenging to assess, and there have been few research studies of good quality.

As reported in the previous section, we will highlight and review the most recent published studies that we consider relevant for the field [79,80,81,82,83,84], distinguishing between EVs’ sources—healthy or cancer cells. 

##### Healthy-Cell-Derived EVs

Macrophages—Macrophage-derived EVs have been examined as potential candidates for producing biomimetic NPs. On this occasion, ultrasmall Ag_2_S quantum dots (QDs) and doxorubicin hydrochloride (DOX) were simultaneously encapsulated into RAW 264.7-macrophage-secreted vesicles through electroporation to give rise to MVs@QDs&DOX [82]. After 10 min of NIR laser irradiation (808 nm), the shells of macrophage-derived EVs is altered. Then, QDs and DOX can be released for deep tumor penetration of the QDs, producing a synergistic effect of the chemotherapy. One of the advantages of using macrophage-derived EVs is the presence of specific cancer ligands. For instance, MVs@QDs&DOX displayed that the CD49d protein that can bind to VCAM-1 overexpressed on 4T1 tumor cells. Thus, MVs@QDs&DOX displayed efficient accumulation in the tumor tissues of the breast cancer 4T1-tumor-bearing Balb/c mice. Moreover, tumor penetration of QDs and anti-tumor effects were enhanced by NIR light irradiation, achieving a tumor inhibition rate of 86%. In addition, macrophage-derived EVs favored QDs immune system evasion, and the blood-circulation time of the “camouflaged” NPs was about four-fold higher than that of free QDs.

Mesenchymal stem cells—mesenchymal-stem-cell-derived EVs have also been explored for encapsulating NIR-responsive nanomaterials. For example, Huang and co-workers synthesized gold nanostars (GNS) with the trans-activating transcriptional (TAT) peptide on the NP surface to increase intracellular uptake in MSCs. TAT-GNS NPs were incubated with MSC for 24 h. Then, released GNS clusters were collected from supernatants of GNS-labeled MSCs that displayed the same protein profile as the unlabeled MSC EVs, confirming encapsulation of TAT-GNS NPs in the EVs. Altanerova et al. [81] used a similar strategy to camouflage NIR-responsive materials within MSC-derived exosomes. The authors incubated iron oxide (Venofer) NPs and MSCs expressing the mRNA of a suicide gene (cytosine deaminase: uracil phosphoribosyl transferase fusion gene) together for at least 3 days. The iron-oxide-rich materials were later exocytosed from Venofer-labeled MSCs. Separation of Venofer-exosomes from secretome was then performed by size-exclusion chromatography. Moreover, the authors confirmed that Venofer-MSC-secreted exosomes can be effectively internalized by HeLa and PC3 tumor cells [81].

##### Cancer-Cell-Derived EVs

As discussed in the previous section, healthy-cell-derived EVs can be decorated by several ligands, such as CD49d protein, that specifically bind to cancer cells and enable cancer homing. However, cancer cells show cell self-recognition; as mentioned above, they possess a natural characteristic, homotypic binding, wherein they tend to adhere to each other, leading to the continuous growth of tumors. Thus, it is reasonable to think that tumor-derived EVs could retain the homotypic targeting capacity, becoming an excellent material for cloaking NPs to increase specific tumor targeting [87]. Although of paramount relevance, this approach is very innovative, and few research efforts have been made. In any case, combinatorial therapies are the current choice for cancer-cell-derived EVs for PTT.

In one example, mesoporous silica NPs (E-MSNs) were used as nano vehicles for co-loading ICG and DOX and were camouflaged with 4T1 cell-derived exosomes (ID@E-MSNs) following an extrusion protocol using a polycarbonate membrane. Upon 808 nm NIR irradiation, ICG produced hyperthermia, and DOX was released (Figure 4) [83]. Then, ID@E-MSNs were administered to 4T1-tumor-bearing BALB/c mice, confirming the effective accumulation of NPs in the tumor sites. Tumor growth was inhibited after 16 days of treatment, which points to ID@E-MSNs as an effective chemo-PTT for preventing growth and metastasis of breast cancer.

The combination of PTT with immunotherapy is also being investigated. PTT can induce immunogenic cell death, activating dendritic cells (DCs), in which maturation is induced by tumor-associated antigens generated in dying cancer cells after treatment.

CD47 is a transmembrane protein that mainly functions as an anti-phagocytic or “do not eat me” signal. CD47 directly binds with SIRPα, mainly located on macrophages, enabling CD47-expressing cells to evade immune clearance [88]. CT26 cells were genetically engineered to obtain CD47-overexpressed exosomes, fused with thermosensitive liposomes (TRL), and loaded with ICG and R837 (an immune adjuvant) for producing NPs, named I/R@hGLV [84]. The presence of the exosomes covers endowed I/R@hGLV NPs with the ability to target homologous cells, enhancing cellular uptake. At the same time, ICG plus NIR laser irradiation triggered DCs maturation by exposing tumor-associated antigens present in the generated cell debris with the assisted function of R837. In addition, the treatment with CD47-overexpressed I/R@hGLV NPs blocked the interaction between macrophages and cancer cells through competitively occupying SIRPα in vitro. In vivo, studies using CT26-tumor-bearing mice demonstrated that I/R@hGLV exhibited longer blood circulation time than particles without the exosomal coating and increased accumulation in the tumor site. Regarding the effect of photo-immunotherapy in vivo, the treatment with I/R@hGLV plus NIR irradiation significantly reduced the tumor volume and increased the number of DCs recruited to the tumor site compared to the control group. DCs maturation studies in vivo showed that R837 is essential to effectively induce DCs maturation and achieve the desired combined effect of PTT with immunotherapy. 

### 3.2. Nature-Inspired Nanoparticles for Photothermal Therapy

Over the past decades, nanomedicine has been inspired to prepare nanoparticles by natural pathogen infection mechanisms. Viruses, bacteria, and toxins can efficiently evade the immune system while targeting the host cells by specific interactions between the pathogen and cell membrane receptors. Viruses, bacteria, or toxins can also recognize molecules that cancer cells express. Thus, research on pathogen-inspired nanoparticles in the context of cancer has been growing in recent years due to their potential to target specific cancer biomarkers. Besides the considerable potential of this approach, only a few articles have reported the application of PTT using the nature-inspired nanoparticles strategy so far.

#### 3.2.1. Virus-Based Nanoparticles

##### Virus-like Particles (VLPs)

VLPs are non-replicative and non-infectious protein particles that closely mimic the structure and size of viruses. They comprise self-assembling viral protein subunits or other molecules derived from viruses, which can assemble spontaneously into VLPs without genetic material [89]. VLPs have become an essential tool in vaccine development against viral diseases [90], but are also used in drug delivery, solid tumor targeting, and theranosis [91]. Recently, VLPs have been investigated as potential PTT platforms in cancer treatment. VLPs have been produced using structural proteins derived from various viruses, including human viruses such as immunodeficiency virus (HIV), hepatitis B virus (HBV), or hepatitis C virus (HCV) [91,92]. For example, hepatitis B core protein virus-like particles (HBc VLP) were used to encapsulate the PTA ICG to target glioblastoma in mice [93]. The HBc-VLP vehicles were modified to improve cancer targeting by including the tripeptide Arg-Gly-Asp (RGD). RGD peptide mainly binds to α_v_β_3_ integrin, which is overexpressed in glioblastoma [93].

Bacteriophages and plant-virus-based nanoparticles (VNP) have also been explored [91,92]. ICG was used for PTT in a bacteriophage-Qβ-based VLP. Bacteriophage Qβ is a small RNA virus that infects bacteria, specifically Escherichia coli. VLPs are expressed in bacteria, specifically bacteriophage Qβ, and incorporate RNA during the assembly process, which affects the adaptive immune response [94]. The authors demonstrated that the mild immunogenicity of Qβ, along with the thermal conversion properties of gold nanostructures and the biodegradability of croconium dyes, results in a robust anti-cancer immune response in a triple-negative breast cancer tumor model in BALB/c mice with lung metastasis [94]. 

Regarding VPNs, a recent exciting study developed using the tobacco mosaic virus (TMV) proteins demonstrated the efficacy of VLPs [95]. The researchers loaded a small molecule immunomodulator called toll-like receptor 7 agonist (1V209) onto TMV and coated its surface with PDA. The modified TMV was then used to treat B16F10 dermal melanoma in C57BL/6 mice, resulting in a potent anti-tumor response achieved through a combination of photothermal therapy and immunotherapy [95]. This study highlights the increased efficacy of combining PTT and immunotherapy; they demonstrated the superior efficacy of the PTT-immunotherapeutic combination over laser or immune-based monotherapies (Figure 5).

##### Virus-Shape-Inspired Nanoparticles

Besides using any of the molecules of the virus for improving nanoparticle cell targeting, researchers have mimicked the morphology of the virus for this purpose. Chen et al. used gold [96], while Wang et al. tuned mesoporous silica nanoparticles to make virus-shaped nanocapsules [97]. Both works demonstrated that the rough surface designed for the virus-like shape nanoparticles enhanced adhesion to promote cellular internalization. Regarding PTT, Tian et al. [98] also showed that raspberry-shaped polypyrrole nanoparticles had superior adsorption at 808 nm and a photothermal conversion effect compared to spherical nanoparticles because of their surface roughness, which enhanced light absorption. Immune responses to the nanocomposites were also dependent on their morphology. Zhong et al. [99] designed a nanoparticle (GNR@HPMO@PVMSN) composed of a gold core, hollow periodic mesoporous organosilica shell, and polydopamine-doped virus-like mesoporous silica nanoparticle outer shell [99]. They loaded the nanoparticle with doxorubicin (DOX) and treated the hepatoma cell line 7721 with GNR@HPMO@PVMSN-DOX. They also performed studies in vivo in tumor-bearing mice. Both in vitro and in vivo studies demonstrated that the virus-like shape provided a much better therapeutic effect on the tumor. In line with this work, Li et al., using HeLa cells and HeLa-tumor-bearing mice as study models, reported enhanced cellular internalization and photothermal conversion ability of spike-like surface nanomaterials made of a NIR-I fluorescence probe (IR825), a chemotherapeutic drug (pemetrexed, PEM), and a rare-earth metal ion (Nd(III)) [100]. The virus-inspired core was coated with a PEG shell to prevent immune clearance and prolong systemic circulation. 

Overall, these studies demonstrate the potential of VLPs and virus-shape-inspired nanoparticles as a platform for PTT in cancer treatment. However, more research is needed to develop and optimize these virus-based therapies for clinical use.

#### 3.2.2. Bacteria-Based Nanoparticles

Bacteria-based nanoparticles have many potential applications, including drug delivery, medical imaging, and cancer therapy [101]. Among the different methods for fabricating bacteria-based NPs, we will highlight the membrane coating technique in this review. 

As previously discussed for eukaryote-derived extracellular vesicles, nanocarriers can also be coated with bacterial membrane vesicles (MVs), such as bacterial outer membrane vesicles (OMVs) released by Gram-negative bacteria [101,102] and Gram-positive-released membrane vesicles referred to as cytoplasmic membrane vesicles (CMVs) [102]. MVs share significant similarities with mammalian EVs in structure and biological activities. Bacterial vesicles have been shown to have a range of biological functions, including cell–cell communication, host–pathogen interactions, and immunomodulation [102]. Thus, biomimetic nanoparticles coated with MVs have shown several potential applications, including drug delivery, immunotherapy, and vaccine development. These nanoparticles are designed to mimic the natural properties of MVs, including their ability to interact with immune cells and trigger an immune response. Additionally, using MVs as a coating material can improve the stability and biocompatibility of the nanoparticles, making them more suitable for in vivo use [101,103].

Besides the promising results as delivery nanocarriers, only a few studies have reported the potential of MV-based NPs for PTT in cancer. Liu and colleagues developed a novel strategy using *E. coli* bacterial outer membrane vesicles (OMVs) loaded with iron oxide–manganese oxide composite nanoparticles (Fe_3_O_4_-MnO_2_-OMV NPs, FMO) (Figure 6) [104]. The FMO particles were designed as a sophisticated multimodal therapy for melanoma treatment, comprising chemotherapy, immunotherapy, and photothermal therapy simultaneously. The iron oxide nanoparticles in FMO generated toxic ROS, while the manganese oxide nanoparticles produced Mn^2+^ and O_2_ in the melanoma tumor microenvironment. The Mn^2+^ acted as an immune adjuvant, while O_2_ prevented immunosuppression caused by hypoxia and enhanced the immunotherapeutic effect. The manganese oxide nanoparticles also had excellent photothermal conversion ability, making them suitable for PTT. *E. coli* bacterial OMVs acted as immune adjuvants to promote immunotherapy and facilitated the accumulation of FMO at tumor sites. Notably, the FMO particles were taken up by neutrophils and accumulated at inflammatory tumor sites, triggering the innate immune system and photothermal damage to destroy cancer cells.

Another exciting strategy based on OMVs for treating melanoma was reported by Peng et al. [105]. This study used OMVs as a delivery system for recombinant human-tumor-necrosis-factor-related apoptosis-inducing ligand (TRAIL) in combination with the photosensitizer ICG to induce PTT for triggering the switch of TRAIL-resistant tumor cells into a sensitive state. The OMVs were also modified with a synthesized α_v_β_3_ integrin targeting ligand (RGP) to enhance specific targeting due to α_v_β_3_ integrin being overexpressed in melanoma but not in normal melanocytes. Skin melanoma progression, relapse, and metastasis were effectively delayed through Photo-TRAIL treatment [105].

#### 3.2.3. Cytotoxic-Protein-Based Nanoparticles

Different natural-derived peptides, venom components, and toxins from various sources have been traditionally used as highly potent cytotoxic drugs for cancer treatment. Nevertheless, over the past few years, an innovative approach has emerged using those proteins for precise therapeutics. Besides their intrinsic cytotoxicity, they can be tuned to recruit various clinically interesting functions, such as specific cell-surface receptor binding. Thus, cytotoxic proteins have started to be applicable in targeted cancer nano therapy as biomimetic systems [106].

On the one hand, cytotoxic-protein-based nanoparticles can be engineered to recognize and bind to receptors overexpressed on the surface of cancer cells [106]. For example, conjugate tumor-targeted cytotoxic nanoparticles have been explored by targeting the chemokine receptor 4 (CXCR4), which is overexpressed in more than 23 human cancers [107]. The cationic peptide T22, an effective CXCR4 ligand, was combined with the active fragments of diphtheria toxin [108,109,110,111], pseudomonas aeruginosa exotoxin [109,111], or the plant toxin ricin [112] to create self-assembling toxin-based nanoparticles. These nanoparticles effectively promoted tumor cell death in vitro and potent CXCR4-dependent anti-tumor effects without systemic toxicity in CXCR4^+^ tumor mouse models of lymphoma [108,110], colorectal [111], endometrial cancer [109], and acute myeloid leukemia [112]. Despite these promising results, no studies have been reported yet to include photosensitizers in this approach for PTT and combinatorial treatment.

On the other hand, some cytotoxic proteins have intrinsic selectivity for cancer cells, such as Chlorotoxin (CTX), derived from the venom of the scorpion *Leiurus quinquestriatus*, which has been shown to bind to matrix metalloproteinase-2 (MMP-2) preferentially overexpressed on the surface of neuroectodermal origin [113] and metastatic breast [114] cancer, among others. Different studies reported iron oxide nanoparticles functionalized with CTX to target glioma [115,116,117,118]. Recently, Chung et al. [118] fabricated PEI-coated iron oxide nanoparticles to deliver O6-methylguanine-DNA methyltransferase (MGMT)-targeting siRNA to glioma cells for inhibiting MGMT activity and subsequent reduction of resistance to chemotherapy agents such as Temozolomide (TMZ). They functionalized iron oxide nanoparticles with CTX and the cell-penetrating peptide polyarginine (R10), allowing siRNA to escape endosomal entrapment and degradation. NP-CTX-R10 was able to transfect glioma cells with the siRNA-MGMT, knocking down the gene efficiently and enhancing the sensitivity to TMZ in different glioma cell lines. 

Another example is the case of the Shiga toxin produced by *Shigella dysenteriae* and *Escherichia coli*. This bacterial toxin recognizes the globotriaosylceramide (Gb3/CD77) receptor, which is overexpressed on the cell surface in some cancers such as breast, ovarian, colorectal gastric, head and neck cancer, and prostate, among others [119]. The reported overexpression of Gb3 in many cancers opened, a decade ago, the possibility of using the nontoxic subunit B (StxB) of the Shiga toxin coupled to different materials as contrasting agents for cancer imaging and drugs for chemotherapy and more recently for PTT [120,121,122]. Navarro-Palomares and colleagues reported the fabrication of iron oxide nanoparticle cores coated with amorphous silica shells containing rhodamine isothiocyanate (RBITC) dye (Fe_3_O_4_@SiO_2_:RBITC) and functionalized with the StxB to target head and neck cancer (HNC) [123]. The same research group recently developed a nanocarrier based on StxB for PTT [124]. Gold nanorods (AuNRs) were coated with silica, RBITC, and the StxB (Figure 7). The pharyngeal cancer cells Detroit 562 and human HNC squamous cell carcinoma samples Gb3^+^ were treated with AuNRs@SiO_2_@ShTxB and irradiated with a NIR laser, inducing cancer cell death efficiently. Furthermore, in vivo studies were developed for translational purposes; a squamous cell carcinoma murine model was treated topically by adding AuNRs@SiO_2_:RBTIC@ShTxB to the drinking water. Irradiation of mice tongues at 808 nm destroyed the affected Gb3^+^ neoplastic tissue, supporting the localized photo-induced hyperthermic effect.

## 4. Current Examples of Clinical Applications

As discussed earlier, photothermal therapy (PTT) approaches utilizing biomimetic nanocarriers exhibit significant potential for cancer treatment. However, the majority of studies have been confined to preclinical settings, and the clinical translation of these techniques remains limited. Only a handful of early-phase pilot clinical trials employing “traditional” PTT strategies have been registered on ClinicalTrials.gov (October 2023), primarily investigating Indocyanine Green (ICG) and the AuroLase Therapy developed by Nanoespectra Biosciences (see Table 4).

It is worth noting that ICG is an FDA-approved near-infrared (NIR)-responsive material used as a medical contrast agent for intravenous administration (NDA: 011525). Presently, there are 319 clinical studies registered to evaluate ICG in various cancer applications, with the majority focusing on its role as a tracer for sentinel lymph node detection, fluorescence-guided tumor resection, mapping, and margin identification. The first pilot clinical trial assessing the potential of ICG for PTT in cancer treatment was reported in 2005. It evaluated the combined effect of transpupillary thermotherapy (TTT) and ICG for treating small and medium choroidal melanomas in 25 patients. The results were promising, with all tumors except one showing a significant volume reduction without clinical evidence of recurrences. However, a subsequent retrospective review of medical records of 391 choroidal melanoma patients treated with TTT-ICG from 1995 to 2012 revealed a direct correlation between a higher number of high-risk tumor features and increased tumor recurrence rates following thermotherapy (NCT01253759) [125]. Subsequently, a study utilizing TTT-ICG in conjunction with intravitreal injection of Ranibizumab (Lucentis) was conducted for choroidal melanoma treatment (NCT00680225). Although the recruitment phase is completed, no data on outcomes or adverse events have been reported.

ICG was also investigated in combination with the immunoadjuvant glycated chitosan for the treatment of advanced breast cancer patients [126]. Ten patients were enrolled in this study from September 2009 to August 2010, with two patients withdrawing from the study. The treatment induced several local adverse events, such as redness, pain, edema, and ulceration at the treatment site. Patients who had received radiation therapy prior to PTT experienced severe pain, swelling, and ulceration. Nevertheless, no severe systemic adverse events were observed. In terms of efficacy, the pilot study demonstrated an objective response rate of 62.5% and a clinical benefit response rate of 75% [126].

AuroLase Therapy, developed by Nanoespectra Biosciences, relies on silica–gold nanoshells of approximately 150 nm coated with polyethylene glycol (PEG), known as AuroShells. These nanoshells have previously demonstrated high effectiveness in absorbing NIR light and converting it into heat, leading to selective hyperthermic cancer cell death, as demonstrated in in vitro and preclinical in vivo studies. AuroShells have also exhibited accumulation at tumor sites via the enhanced permeability and retention (EPR) effect following intravenous injection.

The first clinical trial employing AuroShell particles commenced in 2008 to assess their efficacy in treating head and neck tumors in 11 patients (NCT00848042). Participants received a single dose of AuroShell particles, followed by one or more interstitial NIR irradiations using an 808 nm laser for photothermal ablation of target lesions. The trial was structured into three arms, varying the concentrations of nanoparticles (4.5 mL/kg, 7.5 mL/kg) and laser irradiation power (3.5 W, 4.5 W, 5.0 W). The study concluded in 2014, reporting no adverse effects attributed to AuroShell particle administration.

Subsequently, Nanoespectra Biosciences initiated a second clinical study from 2012 to 2014 to evaluate the efficacy of AuroLase therapy in lung tumors. This study included patients with primary and metastatic lung and airway obstruction tumors who received a single dose of particles followed by NIR laser irradiation via bronchoscopy-delivered optical fiber. Unfortunately, the trial was terminated prematurely due to a significant portion of patients not demonstrating thermal lesions (NCT01679470).

The most recent study involving AuroShells was a phase I trial targeting prostate tumors through nanoparticle-directed irradiation (NCT02680535) [127]. AuroLase^®^ was assessed in 16 patients with low- to intermediate-risk localized prostate cancer. Patients received an intravenous infusion of AuroShell particles, followed by transperineal magnetic resonance–ultrasound fusion imaging (MRI/US)-guided placement of an interstitial optical fiber within the tumor, followed by laser treatment with NIR light. Biopsy results showed that ablation zones were cancer-free in 60% (9/15) of patients at 3 months and 86.7% (13/15) at 12 months, with minimal damage to healthy tissue. The success of this pilot study has paved the way for further investigations, including an ongoing extension study (NCT04240639).

## 5. Challenges in Translating from Bench to Bedside

Despite the outstanding properties of some of the materials described in this review and the high promise of PTT as a single or combinatorial treatment against cancer, there are still a few hurdles that need to be overcome for PTT to impact cancer management and patients’ well-being. Some of these challenges are common to all nanotechnology applications in biomedicine. Topics such as batch-to-batch reproducibility, quality control, and the elimination of nanostructures from the body continue to pose challenges for the translation of nanoparticle-based solutions into clinical practice. However, researchers are increasingly considering industry compatibility from the outset, aiming to facilitate scalability. This, coupled with advancements in synthetic protocols and the availability of fabrication and characterization instrumentation, is expected to alleviate short-term concerns regarding batch quality.

The issue of nanostructure elimination from the body appears more complex to resolve. Each material type and functionalization exhibit unique pharmacodynamics and pharmacokinetics, necessitating a case-by-case approach to address this issue. Further research is needed to assess the mid-to-long-term fate of nanostructures in the body. Some materials and structures proposed for PTT have only recently been introduced to biomedicine, and data on their behavior are not yet available. Consequently, alternative materials, even if their performance in PTT is significantly inferior to gold standards, have been explored extensively because they have been in clinical use for several decades (e.g., iron oxides) and more exposure data are accessible.

Another common issue, arguably the most significant, affecting nanostructure-based drugs (nano-drugs) is their limited specificity. Whether nano-drugs are functionalized with molecules (e.g., antibodies) to enhance their specificity toward cancer cells or not, only a minimal proportion of the injected dose reaches the tumor. This limitation seriously hinders the effectiveness of nano-drug-based therapies and increases the likelihood of off-target effects. These secondary effects are often related to the toxicity of the nano-drugs. In the case of PTT, the premise is that PTT agents should be safe unless irradiated, theoretically minimizing off-target effects. However, as with other cancer treatment modalities, monotherapy is typically insufficient to cure the disease. Combining PTT with other treatment modalities can increase the toxicity of PTT agents in the absence of radiation. Further research is required to (i) enhance the specificity of nano-drugs and (ii) improve PTT probes for combinatorial treatment. Additionally, enhancing the PCE of nanomaterials will reduce the laser power required for PTT, subsequently decreasing laser-associated side effects and enhancing overall treatment safety.

In terms of instrumentation, a primary constraint of PTT is the limited penetration of light into biological tissues. So far, melanoma has been the most studied tumor type for PTT applications. While the use of near-infrared (NIR) excitation has somewhat improved the situation, deep-seated tumors present a significant challenge for PTT. In this regard, the development of minimally invasive laser systems for use within the body, similar to current applications in endovenous laser therapy for varicose veins or coronary interventions, could offer a path forward [128]. Additionally, monitoring the temperature of tumors in real-time remains a significant challenge. In superficial tumors, infrared imaging is the gold standard, providing surface temperature data only. Temperature recording technologies are urgently needed to measure the temperature of tumors (both superficial and deep-seated) at various depths along the tumor and longitudinally during PTT treatment. Such developments would benefit PTT and other forms of hyperthermia treatment (magnetic, ultrasounds, radiofrequency, etc.), all of which suffer from a limited understanding of the temperature profile within tumors. Improved instrumentation would enable more detailed treatment planning, ultimately enhancing treatment outcomes and patient prognosis.

Regarding the application of PTT, healthcare professionals, including radiologists and oncologists, will require extensive training on PTT agents and instrumentation to safely administer this type of therapy.

Regulatory matters have also historically posed challenges and raised concerns about nanoparticle-based medical solutions. However, as an increasing number of nano-drugs and nano-devices gain approval from regulatory agencies such as the FDA and EMA, this issue has been mitigated and represents less of a hurdle. Nevertheless, some concepts explored in this review, such as the use of membranes and natural structures (viruses, bacteria, toxins) to cloak or direct these agents to tumors, may face additional delays due to the stringent regulations governing biostructures, which are already subject to heavy regulation even without the nanostructure component (e.g., toxins). However, recent experiences with COVID-19 vaccines have shown that regulatory flexibility can be achieved when the benefits outweigh the potential risks.

## 6. Conclusions and Future Perspectives

Cancer is still one of the most significant challenges of our time. Monotherapies are frequently not the best therapeutic option for cancer patients; beyond the three classic pillars of oncology therapeutics (surgery, chemotherapy, radiotherapy) together with modern immuno-therapy, several other exciting options are being developed to bring hope, mainly in the form of combinatorial treatments. One of these options is PTT. Combining NIR-absorbing materials with NIR lasers can locally increase the temperature of specific tissues (tumors). The effect of this temperature increase can be two-fold.

On the one hand, it can directly damage and promote tumoral cell death. At the same time, on the other hand, it can synergize with other treatments by combining therapeutic outcomes and facilitating treatment access to the tumor (for example, by increasing the tumor permeation of a drug or drug delivery system). The specificity of PTT, even in the absence of tumor targeting in the PTT agent provided by the laser, is one of the strong points of PTT as it allows to spare healthy tissues even if the agent accumulates in those. As described in the section above, a few hurdles must be overcome to translate PTT into the clinic. Still, with the wide choice of materials already described for PTT and the biomimetic strategies compiled in this review for their shielding and homing, there is no doubt that in the mid-term, PTT will find its way into the clinic.

## Figures and Tables

**Figure 1 ijms-24-15484-f001:**
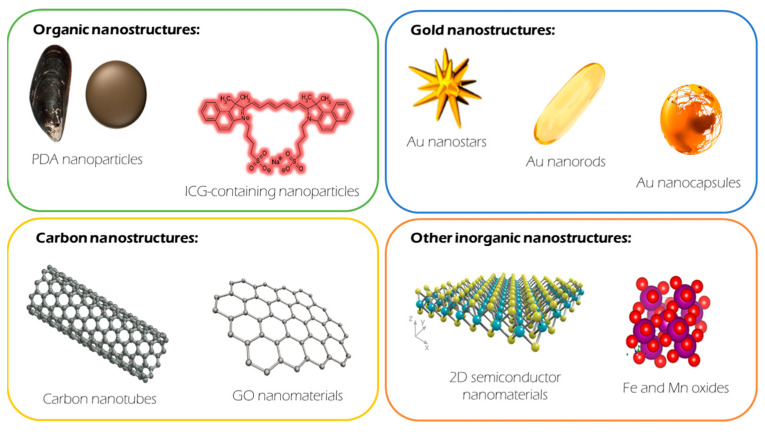
Main families of materials described as photothermal therapy agents and the most representative members within each family.

**Figure 2 ijms-24-15484-f002:**
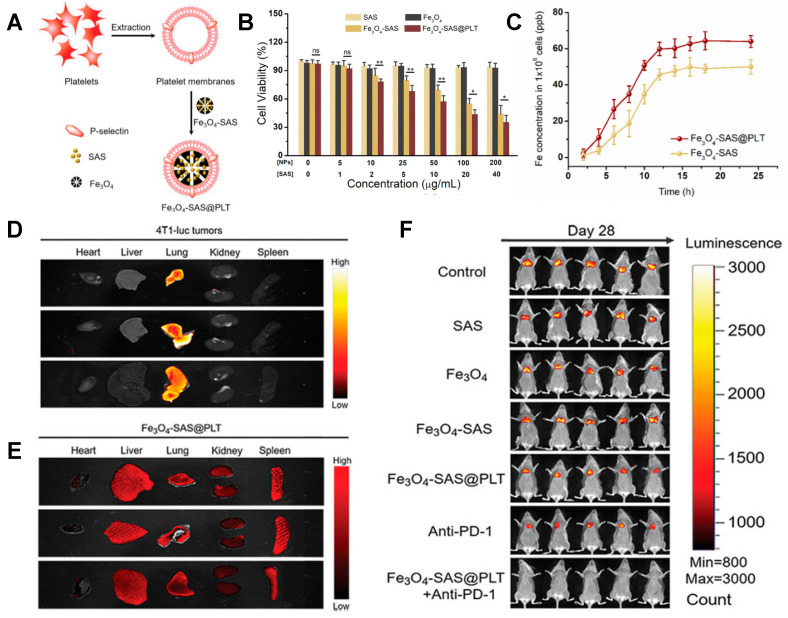
Platelet-membrane-camouflaged iron oxide nanoparticles for ferroptosis-enhanced cancer immunotherapy. (**A**) Schematic illustration of the preparation of Fe_3_O_4_-SAS@PLT. (**B**) Cell viability of 4T1 cells treated with different concentrations of free SAS, Fe_3_O_4_, Fe_3_O_4_-SAS, and Fe_3_O_4_-SAS@PLT, respectively. ns = not statistically significant. * *p* < 0.05, ** *p* < 0.01. (**C**) Cellular uptake of Fe_3_O_4_-SAS@PLT and Fe_3_O_4_-SAS nanoparticles at various incubation times. (**D**) Ex vivo bioluminescence imaging and (**E**) fluorescence imaging of significant organs 24 h post-injection of Fe_3_O_4_-SAS@PLT. (**F**) In vivo bioluminescence images of mice in different groups. Adapted from Jiang et al. Small 2020, 16, 2001704. © 2020 Wiley [54].

**Figure 3 ijms-24-15484-f003:**
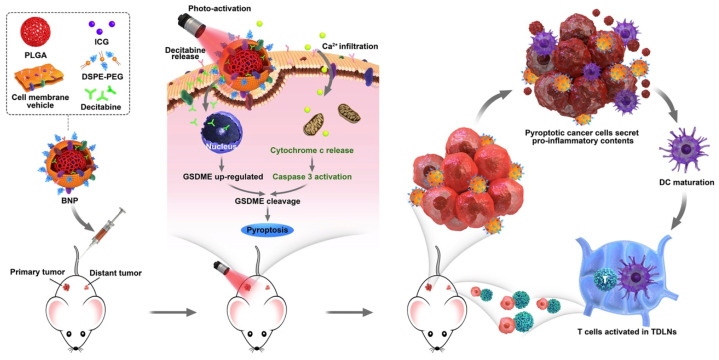
Programming cell pyroptosis with biomimetic nanoparticles. Tumor-homing nanoparticles (BNP) can effectively accumulate in primary tumors when administered intravenously. These nanoparticles can induce cancer cell pyroptosis, which causes the cancer cells to secrete pro-inflammatory contents that trigger dendritic cell maturation and T-cell activation in the tumor-draining lymph nodes. This process can lead to primary and remote tumor immunotherapy. Reprinted with permission from Zhao et al. Biomaterials, 2020. 254:120142. © 2020 Elsevier [74].

**Figure 4 ijms-24-15484-f004:**
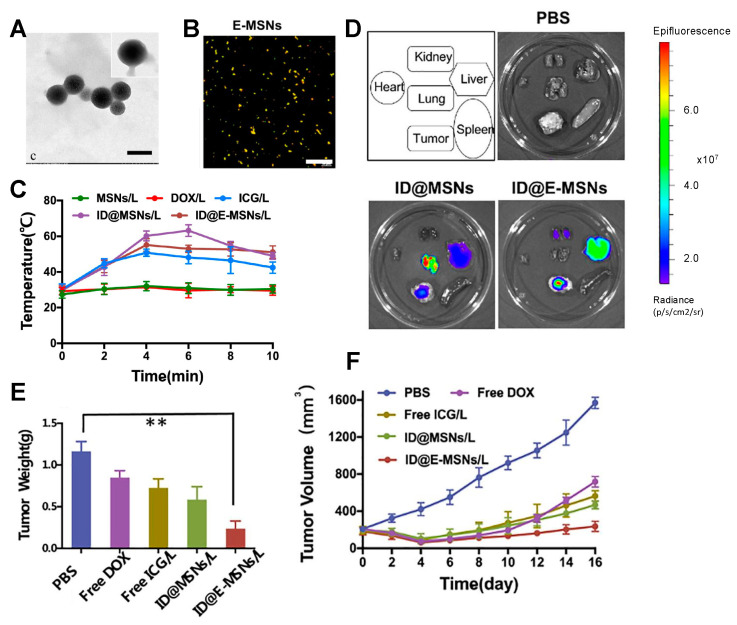
Tumor exosome-mimicking nanoparticles for tumor combinatorial chemo-photothermal therapy. (**A**) TEM images of E-MSNs. (**B**) Confocal images of fused E-MSNs nanovesicles. The exosome membrane is labeled by DiO dye (λ_em_ = 488 nm), DiI loaded in MSNs (λ_em_ = 560 nm). (**C**) Temperature curves of laser irradiated PBS, free DOX, free ICG, ID@MSNs and ID@E-MSNs. (**D**) Fluorescence images of organs harvested from 4T1 tumor-bearing mice that were injected with ID@MSNs and ID@E-MSNs after 24 h. (**E**) Tumor weight, n = 5. Data are characterized as mean ± SD, ** *p* < 0.01 (**F**) Tumor volumes. Adapted from CI Tian et al. N Front. Bioeng. Biotechnol. (2020) 8:1010. Creative Commons CC-BY license [83].

**Figure 5 ijms-24-15484-f005:**
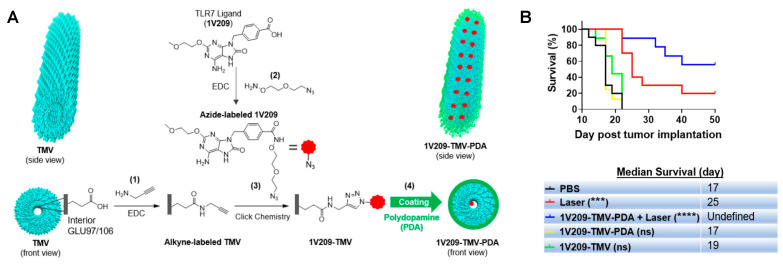
Photothermal immunotherapy of melanoma using the immunomodulator toll-like receptor 7 agonist (1V209) laden tobacco mosaic virus with a polydopamine coat. (**A**) Schematic of bioconjugation and coating reactions for synthesizing 1V209-TMV-PDA. (**B**) Survival data of TMV-based PTT treated melanoma-bearing mice. PTT showing greater efficacy when combined with immunostimulatory nanoparticles (1V209-TMV-PDA), indicating that 1V209-TMV-PDA + laser irradiation outperformed all other treatment groups (with ˃50% mice showing median survival ˃ 50 days). Asterisks indicate significant difference between a given group and PBS (control) (*** *p* < 0.001; **** *p* < 0.0001). Adapted from C.I. Nkanga et al. Nanomedicine: Nanotechnology, Biology, and Medicine 44 (2022) 102573 © 2020 Elsevier [95].

**Figure 6 ijms-24-15484-f006:**
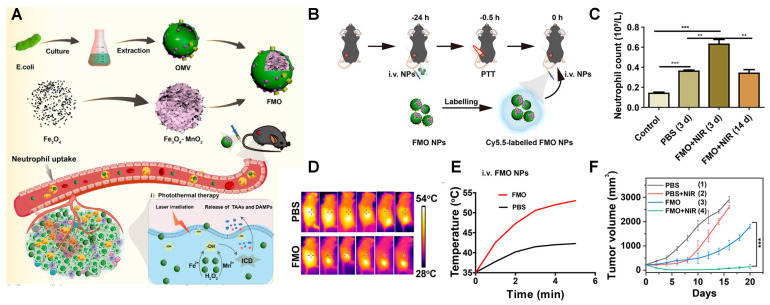
Bacteria-derived outer membrane vesicle (OMV)-functionalized Fe_3_O_4_-MnO_2_ (FMO) nano platform for neutrophil-mediated targeted delivery and photothermally enhanced cancer immunotherapy. (**A**) Schematic illustration of the synthesis and therapeutic mechanism of OMV camouflaging Fe_3_O_4_-MnO_2_ NPs to generate FMO NPs. (**B**) Experimental process of PTT, enhancing the tumor inflammatory microenvironment and recruiting neutrophils to carry FMO NPs. (**C**) Blood biochemical assays of neutrophil counts in mice before and after PTT treatment. Student’s *t* test was used to analyze statistical differences, where ** *p* < 0.01, and *** *p* < 0.001 were considered statistically significant. (**D**) Infrared thermographic images of mice in PBS + NIR and FMO + NIR groups. (**E**) Corresponding temperature curves of mice. (**F**) Tumor volume changes in mice during treatments. Adapted with permission from ACS Appl. Mater. Interfaces 2023, 15, 3, 3744–3759. Copyright 2023. American Chemical Society [104].

**Figure 7 ijms-24-15484-f007:**
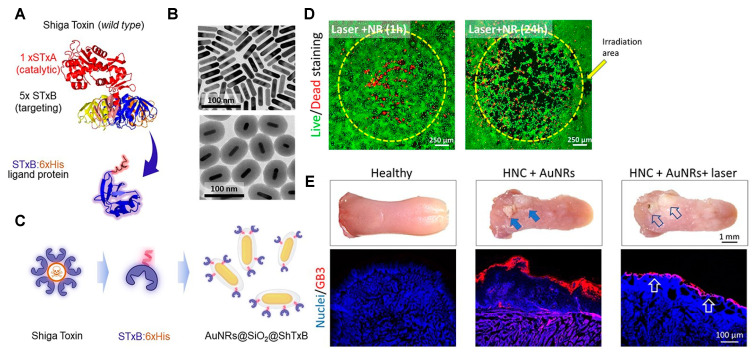
Photothermal therapy for squamous cell carcinoma using AuNRs@SiO_2_ coated with the Shiga toxin B-subunit. (**A**) Shiga toxin (ShTx) structure. (**B**) TEM images of AuNRs before and after SiO_2_ coating. (**C**) Shiga toxin B-subunit (ShTxB) (blue) is attached to the 6xHis cationic tail (red) for production of AuNRs@SiO_2_@ShTxB. (**D**) Live (green)/dead (red) staining of Gb3^+^ HNC cells treated with the AuNRs@SiO_2_@ShTxB and exposed to PTT. (**E**) Post-mortem murine tongues. Blue solid arrows point to representative carcinoma lesions. Empty blue arrows show the same areas after irradiation with an 808 nm LD. Adapted from Navarro-Palomares, E. IJN Volume 17, 5747–5760 (2022). © DovePress (CC BY-NC 3.0) [124].

**Table 1 ijms-24-15484-t001:** Healthy cell-membrane-cloaked nanomaterials developed for PTT in cancer.

Shell (Cell Membrane)	Core (NIR-Responsive Materials)	Therapy	Cancer	Study Models	Reference
Sheep Red blood cells (RBC)	Titanium dioxide/gold nanorods	PTT	Human breast cancer	In vitro: MCF7 cells	[43]
Mouse RBC	Gold nanocages		Murine breast cancer	In vitro: HepG2 cells; In vivo: BALB/c mouse bearing 4T1	[44]
	Iron oxide nanoclusters		Human breast cancer	In vitro: MCF-7 cells; In vivo: Nude mice bearing MCF-7	[45]
	Iron oxide nanoclusters + Cypate		Human colon cancer	In vitro: HCT-116 cells; In vivo: Balb/c nude mice bearing HCT-116	[46]
	BSA + indocyanine green (ICG) + gambogic acid	Chemo-PTT	Human breast and cervical cancer	In vitro: HeLa and MCF7 cells; In vivo: Nude mice bearing HeLa	[47]
Mouse RBC + anti-EpCam	Gold nanocages + paclitaxel		Murine breast cancer	In vitro: 4T1 cancer cells	[48]
Mouse or human RBC + Folic acid	Graphene oxide + ICG + doxorubicin (DOX)		Human cervical cancer	In vitro: HeLa cells; In vivo: BALB/c mice bearing HeLa	[49]
Mouse or human RBC + Hyaluronic acid	Prussian blue + gamabufotalin		Human breast cancer	In vitro: MDA-MB-231 cells; In vivo: BALB/c mice bearing MDA-MB-231	[50]
Mouse Platelets (PLT)	Bismuth-silica nanorods	Radio-PTT	Murine breast cancer	In vitro: 4T1 cancer cells; In vivo: BALB/c mice bearing 4T1	[51]
	Polypyrrole + DOX	Chemo-PTT	Human hepatocellular carcinoma	In vitro: Huh7 cells; In vivo: BALB/c nude mice bearing Huh7	[52]
	PLGA + IR780 + DOX		Murine breast cancer	In vitro: 4T1 cancer cells; In vivo: BALB/c mice bearing 4T1	[53]
	Iron oxide + sulfasalazine	Chemo-PTT + immunotherapy	Murine breast cancer	In vitro: 4T1 cancer cells; In vivo: BALB/c mice bearing 4T1	[54]
Mouse Macrophage (Ma)	Iron oxide nanoparticles	PTT	Human breast cancer	In vitro: MCF7 cells; In vivo: BALB/c nude mice bearing MCF-7	[55]
	Gold–silica nanoshells		Murine breast cancer	In vitro: 4T1 cancer cells; In vivo: BALB/c mice bearing 4T1	[56]
Rat alveolar Ma			Murine glioma	In vitro: C6 glioma cells; In vivo: Sprague-Dawley rat glioma	[57]
Human umbilical cord mesenchymal stem cell (UCMS)	polydopamine (PDA) nanoparticles + SN38	Chemo-PTT	Human osteosarcoma	In vitro: MG63 cancer cells; In vivo: BALB/c mice bearing MG63	[58]
	PDA—iron oxide nanoparticles + Plk1 siRNA	Gene-PTT	Human prostate cancer	In vitro: DU145 cancer cells; In vivo: Nude mice bearing DU145	[59]

**Table 2 ijms-24-15484-t002:** Cancer cell membrane-cloaked nanomaterials developed for PTT in cancer.

Shell (Cell Membrane)	Core (NIR-Responsive Materials)	Therapy	Cancer Target	Study Models	Reference
Human hepatocellular carcinoma (HepG2 cells)	Iron oxide @ silicon dioxide	PTT	Human hepatocellular carcinoma Human melanoma	In vitro: HepG2 cells In vitro: A375 cells	[68]
Human lung cancer (A549 cells)	PLGA @ ICG + perfluorocarbons		Human lung cancer	In vitro: A549 cells In vivo: Nude mice bearing A549	[69]
Human breast cancer (MCF7 cells)/ Mouse RBC	Melanin		Human breast cancer	In vitro: MCF-7 cells In vivo: Nude mice bearing MCF7	[70]
Human hepatocellular carcinoma (HepG2 cells)	Liposome@ICG + DOXO	Chemo-PTT	Human hepatocellular carcinoma	In vitro: HepG2 cells In vivo: BALB/c nude mice bearing HepG2	[71]
Mouse melanoma (B16-F10 cells)	Hollow copper sulfide @ ICG + DOXO		Mouse melanoma	In vitro: B16-F10 cells In vivo: C57BL/6 mice bearing B16-F10	[72]
Mouse hepatocellular carcinoma (H22 cells)/ Mouse macrophage (RAW 264.7 cells) + anti-VEGFR	Hollow copper sulfide @ Sorafenib		Human and murine hepatocellular carcinoma	In vitro: HepG2 cells In vivo: BALB/c mice bearing H22	[73]
Mouse breast cancer (4T1 cells)	PLGA @ ICG + decitabine	Chemo-PTT + immunotherapy	Murine breast cancer	In vitro: 4T1 cells In vivo: BALB/c mice bearing 4T1	[74]
Mouse melanoma (B16-F10 cells)/ bacterial outer membrane	Hollow polydopamine	PTT + immunotherapy	Human lung carcinoma; Human breast cancer; Mouse melanoma	In vitro: A549, MCF7 cells In vivo: C57BL/6 mice bearing B16-F10	[75]

**Table 3 ijms-24-15484-t003:** EV-cloaked nanomaterials developed for PTT in cancer.

Shell (Extracellular Vesicles)		Core (NIR-Responsive Materials)	Therapy	Cancer	Study Models	Reference
Healthy cell-derived EVs	Mouse adipose stem cells	Iron oxide (USPIO)	MRI			[79]
	Human umbilical cord mesenchymal stem cell	Gold nanostars -TAT peptide	PTT	Human prostate cancer	In vitro: PC-3 cells; In vivo: Nude mice bearing PC-3	[80]
	*yCD::UPRT* fusion gene human mesenchymal stem cells	iron oxide (Venofer)	Hyperthermia	Human prostate and breast cancer	In vitro: PC-3 cells and HeLa cells	[81]
	Mouse macrophage (RAW 264.7 cells)	Ag2S quantum dots @ DOX	Chemo-PTT	Mouse breast cancer	In vitro: 4T1 cells; In vivo: Balb/c mice bearing 4T1	[82]
Cancer cell-derived EVs	Mouse breast cancer (4T1 cells)	Mesoporous silica @ ICG + DOX				[83]
	Mouse colon cancer (CD47-overexpressed CT26 cells)	Liposomes @ ICG + R837	Immuno-PTT	Mouse colorectal carcinoma	In vitro: CT26 cells; In vivo: BALB/C mice bearing CT26	[84]

**Table 4 ijms-24-15484-t004:** Clinical trials of NIR-responsive materials for cancer PTT.

NIR-Responsive Material	Cancer Type	Clinical Trial	Recruitment Status
Indocyanine green (ICG)	Choroidal melanomas	NCT01253759	Completed
ICG + Ranibizumab		NCT00680225	Completed
Silica–gold nanoshells (AuroShell^®^)	Head and neck	NCT00848042	Completed
	Lung tumors	NCT01679470	Terminated
	Prostate	NCT02680535	Completed
		NCT04240639	Active, not recruiting

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
