# Peer review of "Recent Advances in Biomimetic Nanocarrier-Based Photothermal Therapy for Cancer Treatment"

_ijms, 2023, doi:10.3390/ijms242015484_

Round 1
Reviewer 1 Report
The article summarises the current knowledge on the photothermal cancer therapy. The article is very comprehensive and informative. A small drawback of the subject is that a very similar review appeared recently (Materials & Design, 217 (2022) 110656. This fact may lower the citations for the current article. Nevertheless, the current contribution covers some other aspects of photothermal cancer therapy.
In introduction Authors mention three traditional pillars for cancer therapy, while in conclusions section they name four. Please be be consistent through out the article.
Please state when ClinicalTrials.net was accessed (page 20).
Some of the statements are not supported by references, see e.g. a paragraph "The work published by Jiang et al. is an exciting example of using PLT membrane- 369 camouflaged Fe3O4 NPs to induce ferroptosis..." on page 9. The missing references should be placed in the text.
All abbreviations should be deciphered prior to their appearance in the text (see eg., "PDT" in page 4). Please check.
Reviewer 2 Report
Comments to author
Manuscript Number/id: - ijms-2659348
The manuscript entitled “Recent advances in biomimetic nanocarriers-based photothermal therapy for cancer treatment” is a nice piece of work done by the authors. In this manuscript, the author reports nanomedicine offers innovative cancer treatment solutions, such as photothermal therapy. PTT uses photoactivatable nanoparticles to absorb near-infrared light and generate heat within cells. The success of PTT depends on core properties and surface properties. A new approach involves creating biomimetic nanoparticles that combine biocompatibility with immune evasion mechanisms. This review outlines strategies for designing biomimetic PTT carriers.
The design and execution of this work are excellent. The study's objectives are adequately described, and the findings are effectively addressed. I recommend for publication of the current version of this article.
